# Social Network Analysis on the Mobility of Three Vulnerable Population Subgroups: Domestic Workers, Flight Crews, and Sailors during the COVID-19 Pandemic in Hong Kong

**DOI:** 10.3390/ijerph19137565

**Published:** 2022-06-21

**Authors:** Weijun Yu, Cheryll Alipio, Jia’an Wan, Heran Mane, Quynh C. Nguyen

**Affiliations:** 1Department of Epidemiology and Biostatistics, School of Public Health, University of Maryland, College Park, MD 20742, USA; hmane@umd.edu; 2Walter H. Shorenstein Asia-Pacific Research Center, Freeman Spogli Institute for International Studies, Stanford University, Stanford, CA 94305, USA; calipio@stanford.edu; 3Samuel Curtis Johnson Graduate School of Management, Cornell University, Ithaca, NY 14850, USA; jw2282@cornell.edu

**Keywords:** social network analysis, COVID-19, SARS-CoV-2, vulnerable population, domestic worker, flight crew, sailor, mobility, exponential random graph model

## Abstract

Background: Domestic workers, flight crews, and sailors are three vulnerable population subgroups who were required to travel due to occupational demand in Hong Kong during the COVID-19 pandemic. Objective: The aim of this study was to explore the social networks among three vulnerable population subgroups and capture temporal changes in their probability of being exposed to SARS-CoV-2 via mobility. Methods: We included 652 COVID-19 cases and utilized Exponential Random Graph Models to build six social networks: one for the cross-sectional cohort, and five for the temporal wave cohorts, respectively. Vertices were the three vulnerable population subgroups. Edges were shared scenarios where vertices were exposed to SARS-CoV-2. Results: The probability of being exposed to a COVID-19 case in Hong Kong among the three vulnerable population subgroups increased from 3.38% in early 2020 to 5.78% in early 2022. While domestic workers were less mobile intercontinentally compared to flight crews and sailors, domestic workers were 1.81-times in general more likely to be exposed to SARS-CoV-2. Conclusions: Vulnerable populations with similar ages and occupations, especially younger domestic workers and flight crew members, were more likely to be exposed to SARS-CoV-2. Social network analysis can be used to provide critical information on the health risks of infectious diseases to vulnerable populations.

## 1. Introduction

Vulnerability is the state in which a person is exposed to the possibility of physical or emotional harm and cannot protect himself or herself from the harm [1]. Vulnerable populations are groups that experience vulnerability and are not limited to groups with individuals of a specific ethnicity, age, socioeconomic status, sexual orientation, or medical condition [2]. The Code of Federal Regulations specifies additional enhanced protections if vulnerable groups are included in research [3]. Homeless populations, prisoners, and those with substance abuse disorders were studied as vulnerable populations in previous public health studies [4]. Mobility, also referred to as population mobility, is the movement of individuals from one geographic location to another [5]. Mobility is considered to be strongly related to airborne infectious disease [6]. In previous pandemics, such as the severe acute respiratory syndrome (SARS) and influenza A, population mobility via public transportation, such as trains and airplanes, in order to travel from place to place, was found to contribute to the spread of SARS [7] and H1N1 virus [8].

The coronavirus disease (COVID-19) pandemic started in mid-December 2019 [9]. The first case infected by SARS-CoV-2 was reported in Wuhan, China [10]. At the early stage of the COVID-19 pandemic, massive population mobility during the Chinese New Year in 2020 was found to exacerbate the rapid transmission of SARS-CoV-2 in China [11]. COVID-19 related public health restrictions such as lockdowns, heavily impacted certain mobile populations, such as migrant domestic workers, and made them vulnerable [12]. The female-dominated domestic worker labor force faced limited access to resources during the COVID-19 pandemic [13,14]. While they have been considered “essential workers” worldwide and even heroes [15], the pandemic aggravated pre-existing issues, such as low salaries, poor living conditions, and unstable employment [16,17,18]. On the other hand, travel restrictions led to a 43% reduction in passenger volume in 2020 than that in 2019 [19]. This low demand increased layoffs among flight crews [20,21]. Additionally, even though safe flying strategies to contain the spread of SARS-CoV-2 in cabins had been designed and implemented by the International Air Transportation Association in 2020 [22] and high-efficient filtering techniques had been used in aircrafts [19], flight crews were still vulnerable to the increasing risk of SARS-CoV-2 infection and mental health burdens during work [23,24], as well as employment uncertainty [21,25]. Sailors, another vulnerable population during the COVID-19 pandemic who also needed to travel due to occupational demand, were constantly gathered in confined working spaces in which SARS-CoV-2 could spread quickly [26,27]. During the COVID-19 pandemic, deciding which populations were vulnerable was crucial, since it impacted policy makers in their decisions regarding who to serve [28]. Hence, we considered domestic workers, flight crews, and sailors to be three hard-to-track and understudied vulnerable population subgroups in Hong Kong during the COVID-19 pandemic.

In network science, social network is defined as a network that not only refers to online social media platforms, but also has a broader meaning in which the vertices represent human beings, and the edges stand for shared connections or relationships between people [29]. In public health, social network analysis is utilized to understand these connections or relationships and explore how they impact people’s health behaviors [30]. Hong Kong is a special administrative region of China with one of the most densely populated economic and transportation hubs in the world [31,32]. The Center for Health Protection in the Hong Kong Department of Health had been publishing publicly available de-identifiable and individual-level COVID-19 data since 23 January 2020 [33]. These data enabled us to conduct this social network study on the three hard-to-track vulnerable population subgroups in Hong Kong. 

Social network analysis has been utilized in a wide range of public health research for decades, from human immunodeficiency virus (HIV) infection [34] and syphilis [35] to smoking [36] and substance abuse [37]. Existing COVID-19 studies used social network analysis to identify SARS-CoV-2 variants [38] and to build infection networks to estimate health policy implications [39]. Furthermore, Exponential Random Graph Models (ERGMs) is one of the most powerful and popular models [30,40,41] to explore social networks. In existing public health studies, researchers used ERGMs to reconstruct the social network of HIV key populations [42] to estimate and simulate missing network data for the National Longitudinal Study of Adolescent to Adult Health (Add Health) [43] among 14 schools in the United States [44]; explore the association between saliva hormones (cortisol and testosterone) and the social network structure of students from a competitive nursing program [45]; and model a location-based social network to investigate multidrug-resistant tuberculosis [46]. Moreover, ERGMs were utilized to investigate mobility patterns among nomadic herders to help combat emerging diseases [47] and examine homophily effects on demographic age and symptoms among COVID-19 patients in Japan [48]. However, no existing studies have examined social networks of the mobility of domestic workers, flight crews, and sailors during the COVID-19 pandemic.

In this study, we utilized publicly available COVID-19 case daily reports and contact tracing records from the Center for Health Protection in Hong Kong Department of Health for domestic workers, flight crews, sailors, and their close contacts. Our study aims include exploring social networks among the three vulnerable population subgroups and capturing temporal changes in their probability of being exposed to SARS-CoV-2 via mobility. We hypothesize that vulnerable populations with increasing mobility will be at a higher risk of exposure to SARS-CoV-2. This research fills a significant knowledge gap on the social networks of vulnerable populations and their mobility in Hong Kong during the COVID-19 pandemic. Our findings can be utilized to understand SARS-CoV-2 exposure via social networks and risk among vulnerable populations, which could contribute to inform comprehensive public health policies and provide necessary health resources to protect vulnerable populations from bearing disproportionate health risks during the COVID-19 pandemic.

## 2. Materials and Methods

We obtained COVID-19 case reports of 13,400 confirmed COVID-19 cases in Hong Kong from 23 January 2020 to 24 January 2022. Specifically, we retrieved daily situation reports of the COVID-19 cases published by the Center for Health Protection in the Hong Kong Department of Health [49]. We then collected contact tracing reports of each COVID-19 case from vote4.hk [50]. All the data were publicly available and de-identifiable. Each COVID-19 case was assigned a unique tracking ID by the local health authority to protect individual identities while making available valuable information for public health research.

This study was designed to measure each vulnerable population’s social networks in relation to mobility in Hong Kong during the COVID-19 pandemic. We manually filtered 652 cases out of the 13,400 cases pool by reviewing all the situation reports and contact tracing records. The 652 cases in this study met our definition of inclusion in one of the following vulnerable populations during the COVID-19 pandemic: (1) domestic workers, (2) flight crews, and (3) sailors. We included these three vulnerable population subgroups and their reported close contacts as the study population in this study. We stratified the study population into five cohorts by COVID-19 temporal waves in Hong Kong (see Table 1) based on the historical epidemic curve of confirmed COVID-19 cases in Hong Kong (retrieved by 24 January 2022), as published by the Center for Health Protection in the Hong Kong Department of Health [49]. We included the three subgroups because we found that they had similar mobility during the COVID-19 pandemic in Hong Kong, which is a crucial component in constructing the social networks in this study. Specifically, they shared the same destination, Hong Kong, for their temporary or long-term mobility. Moreover, our observed networks demonstrated that these three subgroups were physically present in common scenarios such as the Hong Kong International Airport or flight cabins where they were exposed to SARS-CoV-2.

We built six observed empirical networks for this study: one for the cross-sectional overall cohort, and five for the temporal wave cohorts, respectively. All six networks were undirected and unweighted. A social network is composed of two components: vertices and edges [30]. Vertices were the study population in this study. One vertex stood for one individual. Edges were shared scenarios where vertices were exposed to SARS-CoV-2. If two vertices were physically presented in at least one of the eight scenarios (see Table 2), they were connected by an edge. This process is referred to as “edges formation” in this study. We crosschecked the information and created all the eight scenarios of each vertex from the publicly available contact tracing records [50]. We then manually created edge lists to build each network. Edges were unweighted in this study. The lengths of the edges were not Euclidean distance, thus long and short edges were undifferentiated from each other. We observed edges formation in this study in a binary manner: the presence of a connected edge or not.

Descriptive statistical analysis was conducted to examine the demographic characteristics of the study population. We calculated and reported the mean value for continuous variables (i.e., age) and proportion for categorical variables (i.e., gender). For descriptive statistical analysis on social networks, we computed and reported average density as well as three measures of centrality: degree centrality, betweenness centrality, and eigenvector centrality. Density represents the proportion of observed edges to the maximum of possible edges in a network [29,51]. A density closer to 1 indicates a highly connected network [29]. Degree centrality is the degree of a vertex, or, in other words, the number of edges connected to that vertex [29]. Degree can range from 0 to a finite number. Betweenness centrality is the frequency that a vertex lives on the shortest path that connects other vertices [29,30]. Betweenness centrality measures the extent to which a vertex sits on the short path between other vertices [29,52]. Eigenvector centrality is the degree in which centrality scores take into account the vertices of the vertex’s neighbor [29]. Eigenvector centrality measures how central a vertex’s neighbors are in a network [29,41]. In addition to density and measures of centrality, we also reported network size and number of isolates. Network size is the number of vertices in a network [29,51], while isolates are vertices with a degree of zero that have no edges connected to other vertices [41]. In our study, isolates were included in the network analysis and excluded from network visualization.

We utilized Exponential Random Graph Models (ERGMs) to build and examine our networks. Firstly, we built a null model for the cumulative cross-sectional network with one term, the edges. This null model is a random graph model that shares the same size of vertices as our observed empirical cross-sectional network. We used this null model as a baseline model to examine how the fit models improved by comparing their Akaike Information Criterion (AIC) values with the AIC in the baseline model. The smaller AIC value the better. We then fitted models by adding terms on four vertex attributes: (1) age (continuous variable in years); (2) gender (binary variable: female or male); (3) COVID-19 symptomatic status prior to diagnoses (categorical variable: asymptomatic, symptomatic, or unknown); and (4) vulnerable population category (categorical variable: domestic workers, flight crews, sailors, or close contacts). Monte Carlo Markov Chain (MCMC) maximum likelihood estimation (MLE) was conducted for each fit model. We selected the fit model with the lowest AIC value as our best-fit-model for the cross-sectional network. We extracted the parameter estimates from the best-fit-model and carried out logistic transformation to estimate the probability of edge formation. We also obtained odds ratios through the exponentiation of the parameter estimates of interest. To capture the temporal changes in the probability of edge formation, we built null and fit ERGMs for each of the five waves and their subnetworks, respectively. Subsequently, we ran MCMC MLE and selected the best-fit-models for each of the five subnetworks.

A total of 1000 random networks were simulated for each of the six observed empirical networks based on the best-fit-models. In total, we simulated 6000 networks. We plotted randomly selected simulated networks to visually compare them with the observed empirical networks. In order to examine the quality of our model fit, we conducted goodness-of-fit examination and reported the Monte Carlo empirical *p*-values on each of our predictors. A Monte Carlo empirical *p*-value less than 0.05 indicates poor model fit [41]. After examining model fit, we tested the main effects of the four vertex attributes (age, gender, COVID-19 symptomatic status, and vulnerable population category) on likelihood of edge formation. For instance, are years of age significantly associated with the likelihood of two vertices being connected by an edge? Moreover, we examined homophily effects on the four vertex attributes. Homophily is a principle in social networks which suggests that similar characteristics connect people together [53]. Specifically, we evaluated whether edges are more likely to form between vertices which are similar to each other on the four respective attributes. For example, are two female vertices more likely to be connected by an edge than one female vertex and one male vertex? The results from this examination were interpreted as probability of edges formation. A probability closer to 0 indicates a low possibility of edges formation, while a probability closer to 1 indicates a high possibility of edges formation. We also provided interpretation with odds ratios for some selected results. An odds ratio greater than one indicates that edges formation is more likely to happen in our study population subgroup.

The mobility of our study population was captured based on publicly available contact tracing reports [50]. Temporary and long-term mobility records were included in our analysis. In this study, we defined temporary mobility as travel to work from residence, travel to local social gathering events, and air travel due to occupational demand. We measured our study population’s temporary and long-term mobility through two different methods. The first method was to record shared scenarios where they were physically presented during mobility. We created edges of social networks in this study based on these shared scenarios (Table 2). The second method was to compare our study population’s mobility within Asia and intercontinental mobility by frequency counting. For instance, if a flight crew traveled to Germany and Thailand 14 days before arriving in Hong Kong, we counted “1” for intercontinental mobility and “1” for mobility within Asia for the flight crew subgroup. If a domestic worker traveled to Hong Kong from the Philippines, we counted “0” for intercontinental mobility and “1” for mobility within Asia for the domestic worker subgroup. We then cross-sectionally and temporally compared the differences in frequencies on mobility within Asia and intercontinental mobility among the three vulnerable population subgroups and their close contacts. We did not count mobility within Hong Kong in the comparison analysis because all members of the study population physically presented there.

We evaluated statistical significance in this study at α = 0.05. Network and statistical analyses in this study were performed using R Statistical Software (version 4.1.3; R Foundation for Statistical Computing, Vienna Austria) and SAS^®^ studio version (SAS Institute Inc., Cary, NC, USA), respectively.

## 3. Results

### 3.1. Descriptive Characteristics

Descriptive statistics for the demographic characteristics of our study population are shown in Table 3. Among the study population (N = 652) included in this study, mean age was 37.42 (±11.41) years old, which was younger than the mean age of the general COVID-19 population in Hong Kong (43.62 ± 19.39 years old). The youngest (1-year-old) and the eldest (94-year-old) cases in this study were close contacts of domestic workers. In total, 64.47% of the study population were female, which was higher than the proportion of females in the general COVID-19 population in Hong Kong (51.89%). For COVID-19 symptomatic status prior to diagnoses, 65.64% of the study population was asymptomatic, and only 2.6% of the study population’s COVID-19 symptomatic status was unknown. In our study population, 28.37% were non-Hong Kong residents, which was higher than the proportion of non-Hong Kong residents (4.99%) in the general COVID-19 population. All the domestic workers in this study were labeled as Hong Kong residents by the local health authority, while just 24.17% of flight crews and 10.48% of sailors were Hong Kong residents. The cumulative mortality rate was 0.3% among our study population, which was exceptionally low in comparison to the cumulative mortality rate of 1.59% among the general COVID-19 population in Hong Kong. Specifically, there were only two deceased cases in our study (one sailor and one close contact), which accounted for 0.16% among the vulnerable population and 3% among their close contacts.

For the distribution of the vulnerable population category, 54.5% of the study population were domestic workers (n = 355), 18.4% were flight crews (n = 120), 16.1% were sailors (n = 105), and 11% were close contacts (n = 72) of these three vulnerable population subgroups. For temporal changes in the distribution of the vulnerable population category, the proportion of domestic workers increased dramatically from 19% in wave 1 to 77% in wave 4 and 67% in early wave 5. The proportions of flight crews and sailors fluctuated across the five waves. The proportion of close contacts of the three vulnerable population subgroups dropped sharply from 60% in wave 1 to 1% in early wave 5.

Descriptive statistics for network characteristics are presented in Table 4. Among the five subnetworks, early wave 5 network was the densest (average density = 5.78%), with the highest degree centrality of 22.08 and betweenness centrality of 43.39 among all waves. The wave 3 network indicated the highest average eigenvector centrality of 5.8% among all waves. The correlation between degree centrality and betweenness centrality was positively strong in wave 1 (*r* = 0.775). Degree and eigenvector centrality indicated strong positive correlations in wave 3, wave 4, early wave 5, and cross-sectional overall networks (wave 3 *r* = 0.67; wave 4 *r* = 0.809; early wave 5 *r* = 0.815; cross-sectional *r* = 0.805). However, no strong correlations among centralities were found in the wave 2 network.

### 3.2. Network Visualization

Figure 1 shows the network topology of the six observed empirical networks and six simulated networks. For better network visualization, we excluded isolates (vertices that are not connected to any others) in Figure 1. There were 157 isolated vertices in the cross-sectional network: nine in the wave 1 network, 64 in the wave 2 network, 11 in the wave 3 network, 36 in the wave 4 network, and 38 in the early wave 5 network (Table 4). We color-coded the included vertices into four categories: green vertices are shown as close contacts of the vulnerable population, orange as domestic workers, blue as flight crews, and pink as sailors. The color-coded six observed networks are empirical networks that we created from the observed network data, which are shown on the left-hand side of Figure 1A–F. One simulated network is displayed for each observed network on the right-hand side, also color-coded. We randomly selected the 1000th simulated networks for visualization comparison. All the simulated networks shared the same network size (i.e., number of vertices) and density with the observed networks. However, all the edges in the simulated networks look quite different from the edges in the observed networks. The simulated networks are highly connected, while the observed networks are sparsely connected. The simulated networks for the observed cross-sectional network look similar to the simulated one for the early wave 5 network. In contrast, the simulated networks for the wave 1 and wave 3 networks look alike, while the simulated networks for wave 2 and wave 4 have a resemblance. This can be explained by the similar network size and density between these networks (Table 4).

We observed clusters of connected vertices in each observed network and a large component in the early wave 5 network (Figure 1F). Since the cross-sectional network is the cumulative network of the five subnetworks, we can also observe the large component from the early wave 5 in the cross-sectional network (Figure 1A). Clusters of connected vertices in the wave 1 network occurred among employers that cohabited with domestic workers, flight crews and their close contacts (Figure 1B). Clusters in the wave 2 network showed that domestic workers, flight crews, and sailors can be connected in the same clusters because of international air travel (Figure 1C). Clusters in the wave 3 network were found to have similar patterns as clusters in wave 2, with more vertices categorized as flight crews (Figure 1D). The dominant vertices for clusters in wave 4 were domestic workers (Figure 1E). We also observed clusters that connected flight crews and sailors with domestic workers (Figure 1E). The large component in the early wave 5 network indicated that domestic workers’ international air travel and quarantine accommodations can be connected, which increases their probability of SARS-CoV-2 infection (Figure 1F). A noticeable cluster in the early wave 5 also showed an outbreak among sailors in a ship (Figure 1F).

To further explore the clusters and large component in detail, we zoomed in on each network and displayed the vertices with high degree in Figure 2. The vertex size was weighted by degree. The larger the vertex size, the higher the degree. Each of the vertices were labeled with their COVID-19 symptomatic status prior to diagnoses: “Asymptomatic” or “S” (symptomatic). In the wave 1 network, most of the high-degree vertices were symptomatic, and were close contacts of domestic workers (Figure 2A). In the wave 2 network, we observed that more high-degree vertices were asymptomatic and were of sailors or flight crews (Figure 2B). In wave 3, more asymptomatic high-degree vertices were present, which were mostly of flight crews (Figure 2C). In wave 4, all the two highest-degree clusters were asymptomatic domestic workers (Figure 2D). In the early wave 5 network, the majority of the high-degree clusters were asymptomatic domestic workers, while a few were sailors (Figure 2E). Moreover, the extremely high-degree large component in the early wave 5 network was dominated by asymptomatic domestic workers (Figure 2E). 

### 3.3. Goodness-of-Fit for ERGMs

For model fit, we examined how our models fit the observed network data. The best-fit-models’ terms and statistics are displayed in Table 5. We included one term in all null baseline models, the edges. We added more terms (i.e., predictors) to the best-fit-models. All the AIC values in our best-fit models were reduced from the AIC values in the null baseline models, which indicated that all the predictors that we added to the best-fit-models were useful for the prediction. All the Monte Carlo empirical *p*-values indicated that all the predictors that we added into the best-fit-models were an adequate fit, since all of the *p*-values were greater than 0.05. If the Monte Carlo empirical *p*-value was less than 0.05, it meant that the predictor was a poor fit.

The goodness-of-fit plots for degree distribution of the best-fit-models are presented in Figure 3. There were six panels, with each panel showing the plot for each network. The black lines in the plots indicated degree distribution for our observed empirical network, while the parallel grey lines near the boxplots indicated the variability (95% confidence interval) of degree distribution predictions among our simulated networks. Among the six best-fit-models, the model for the wave 1 network demonstrated the best degree distribution prediction, because the black line in the wave 1 network plot was totally placed inside the parallel grey line band. The model for wave 3 demonstrated fair ability; it overestimated a number of vertices with degree 2. The models for the cross-sectional, wave 2, and wave 4 networks underestimated numbers of isolates and low-degree vertices (degree < 2), and overestimated numbers of some vertices between degrees 5 and 10. The model for the early five wave network underestimated the number of isolates, as well as low-degree (degree < 2) and very high-degree (degree > 20) vertices.

### 3.4. Associations between Vulnerable Population Subgroups’ Attributes and Network Outcomes

We initially examined the basic association between age and vertex degree by plotting a scatterplot (Figure 4). The scatterplot suggests that no linear relationship was found between age and vertex degree. However, our fit ERGMs revealed significant main effects and homophily findings. In the cross-sectional network, age was found to be negatively and significantly associated with the likelihood of two vertices being connected by edges (*p* = 0.046). Working as domestic workers (*p* = 0.009) or flight crews (*p* = 0.003) were positively and significantly associated with the likelihood of edge formation. For homophily effects, being symptomatic prior to a COVID-19 diagnosis increased the likelihood of being connected by an edge with one another (*p* = 0.022). Additionally, if the vulnerable individual fell into the same vulnerable population category, they were more likely to be connected by edges. All of the four vulnerable population categories (domestic worker, flight crew, sailor, and close contact) were significant for homophily effects, respectively (*p* = 0.001).

We then examined each of the five subnetworks for main effects and homophily effects on temporal changes. In the wave 1 network, no significant main effects nor homophily effects were found. In the wave 2 network, working as domestic workers (*p* = 0.004) was positively and significantly associated with the likelihood of edge formation, while working as part of a flight crew (*p* = 0.023) had a negative and significant association. Findings for homophily effects were similar with the cross-sectional network, except working as a domestic worker was not significant in terms of the wave 2 network homophily effect. In the wave 3 network, domestic workers were found to be more likely to be connected by edges (*p* = 0.004). In the wave 4 network, no significant main effects nor homophily effects were found. In the early wave 5 network, age (*p* = 0.001) and working as a sailor (*p* = 0.047) were found to be negatively and significantly associated with the likelihood of two vertices being connected by edges. In contrast, working as a domestic worker (*p* = 0.015) or sailor (*p* = 0.001) were more likely to be connected by an edge with one another.

From January 2020 to January 2022 in Hong Kong, the overall probability that two COVID-19 cases in the study population were connected by an edge was 0.78%. Our study further revealed that the probability of a 30-year-old COVID-19 case and a 50-year-old case with a connected edge was 0.7%, while the probability for a 30-year-old and a 20-year-old would increase to 0.8%. Moreover, the probability that two domestic workers were connected by an edge was 3%, and 2.5% for two flight crews, which were higher than the overall probability of edges formation. Those working as domestic workers were found to be 1.81-times more likely to have edge formation than the general COVID-19 population, in comparison to those working in a flight crew, who were found to be 1.65-times more likely.

For temporal changes in the probability of edges formation, the overall probability of edge formation was 3.38% during wave 1. During wave 2, the overall probability of edge formation decreased to 1.25%, but the probability of two domestic workers with a connected edge was 14.97%. In other words, working as a domestic worker was found to be 4.56-times more likely to have edge formation in wave 2. Overall, the probability of edge formation increased to 4.83% during wave 3 and dropped to 2.06% during wave 4. During the early wave 5, the overall probability of edge formation doubled to 5.78%.

### 3.5. Mobility within Asia and Intercontinentally

The three vulnerable population subgroups indicated different patterns in cumulative cross-sectional mobility from January 2020 to January 2022 (Table 6). Among the 355 domestic workers, 84.5% experienced mobility within Asia, and only 2.82% had intercontinental mobility. Regarding mobility within Asia among domestic workers, mobility was found to show long-term activity. That is, they mainly migrated from Southeast Asian countries, such as the Philippines and Indonesia, to Hong Kong to work. A small portion of intercontinental mobility engaged in by domestic workers was temporary; they travelled from Europe or America to Hong Kong for vacation with or without their employers. For the 120 flight crews, 78.33% had intercontinental mobility around North America, Europe, Africa, and Oceania, and 30.83% were mobile within Asia. The mobility of flight crews was temporary; they travelled around the world mainly for work-related reasons. For sailors, 90.48% were mobile within Asia, while 16.19% engaged in intercontinental mobility. Sailors’ mobility was temporary; they travelled around Asia and Europe for work-related reasons. Regarding the close contacts of the three vulnerable population subgroups, only a few were mobile within Asia (13.89%) and intercontinentally (11.11%). Consequently, their mobility was temporary, and they travelled for personal reasons such as sightseeing. 

We found temporal changes over the five temporal waves in the mobility of the three vulnerable population subgroups within Asia (Figure 5). For mobility within Asia, domestic workers indicated an upwards trend. That is, the proportion of domestic workers who were mobile within Asia almost quadrupled from wave 1 (25%) to early wave 5 (94.53%). Sailors also showed an upwards trend, but this started to decrease after wave 4. Flight crews had a small peak in wave 2 and then their mobility constantly descended. Close contacts of the three vulnerable population subgroups mainly showed a descending trend.

For intercontinental mobility, we also found temporal changes among the three vulnerable population subgroups and their close contacts over the five temporal waves (Figure 6). Flight crews engaged in the most intercontinental mobility among the subgroups; while it fluctuated over the five waves, it maintained a relatively high level. Sailors showed the second highest proportion of intercontinental mobility involvement over wave 2 and wave 4, but this mobility started to descend at wave 4. In contrast, domestic workers experienced a descending trend over the five waves, and rarely engaged in intercontinental mobility following wave 2. Close contacts of the three vulnerable population subgroups also showed a descending trend until wave 4, where it started to increase.

## 4. Discussion

The Hong Kong health authority changed policies regarding the COVID-19 pandemic over the two years during our observation period. As a result, the wave difference was affected by local policy change. Among our study population, gender distribution was found to be skewed with 98.6% of domestic workers being female, while 100% of sailors were male. Furthermore, while most foreign domestic workers originated from outside Hong Kong, 100% of them were categorized as Hong Kong residents by the Hong Kong Department of Health, representing ideal data completeness. Specifically, most domestic workers included in this study originated from Southeast Asian countries, such as the Philippines and Indonesia. While domestic workers were exposed to less mobility intercontinentally but more mobility within Asia compared to flight crews and sailors, domestic workers were 4.56-times in wave 2 and 1.81-times in general more likely to be exposed to SARS-CoV-2 than other vulnerable populations. Flight cabins and public transport were found to be the main risks of infection for newly arrived domestic workers. In contrast, the risk for domestic workers who resided in Hong Kong without recent travel histories came from employers. This is due to Hong Kong’s employment law, which requires domestic workers to “live-in” or cohabitate with their employers, as well as to the high local mobility found among domestics for social gathering activities. Our social networks visualization captured the increasing numbers of clusters and a large component among domestic workers in wave 4 and early wave 5. This indicated that our network structures were capable of estimating the increasing risk among domestic workers.

In comparison, 34% of flight crews and 10% of sailors were Hong Kong residents, indicating higher rates of missing data due to loss of follow-up. Due to the spatial heterogeneity among flight crews and sailors’ places of origin and work sites all over the world, tracking these two vulnerable population subgroups intercontinentally was extremely challenging. Sailors’ places of origin were geographically diverse, ranging from Southeast, South, and West Asia. Additionally, due to their work, there was a lack of follow-up among some sailors who received the viral tests. Likewise, flight crews in this study also indicated diverse places of origin across Asia, Europe, Africa, and North America, and were found to have even higher rates of missing data due to the lack of follow-up.

Despite these challenges with the data, we found that there was an extremely low mortality rate but high rate of asymptomatic cases among our study population, which is suggestive of healthy worker effect [54,55]. Healthy worker effect is a phenomenon that workers have lower mortality rate than the general population because individuals with severe underlying conditions are likely to be excluded from employment [54,55]. The number of reported close contacts of the three vulnerable population subgroups also decreased during our study period, which demonstrates the effectiveness of the local health authority’s public health strategies in containing COVID-19 in local communities before January 2022. However, unlike most Hong Kong residents who were able to increase protection against COVID-19 by reducing mobility, the three vulnerable population subgroups, especially domestic workers, were at a high risk of being infected by SARS-CoV-2 due to the international mobility required by their occupations. The probability of being exposed to a COVID-19 case in Hong Kong among the three vulnerable population subgroups increased from 3.38% in early 2020 to 5.78% in early 2022. On the other hand, we found that having a higher socioeconomic level did not prevent some populations from being vulnerable during the COVID-19 pandemic. Instead, our study indicated that vulnerable populations could have diverse socioeconomic statuses. For example, members of flight crews normally have a higher socioeconomic level than domestic workers and sailors, yet their status does not necessarily change their vulnerability to SARS-CoV-2.

Moreover, in our study, ERGMs were utilized for their ability to manage cross-sectional network data [30,41]. However, the temporal changes in the network dynamic usually cannot be captured by analyzing cross-sectional network data. To compensate for this major limitation of ERGMs, we analyzed both the cumulative cross-sectional network data of this study, and the five subnetworks stratified by the five time periods of COVID-19 waves in Hong Kong. By doing this, we successfully captured unique patterns and changes in terms of the main effects and homophily effects, and probability of edges formation over time for each wave. This innovative methodological design provides a model for future studies that are interested in utilizing the power of ERGMs, but also want to capture the temporal changes in network dynamics. Finally, we used ERGMs to discover findings that otherwise could not be revealed by simple linear regression models, thereby demonstrating the importance of widening the use of social network analysis in future public health studies.

This study, however, has limitations. Firstly, as an observational study, we had no direct access to the individuals of the COVID-19 cases and were not able to determine causal relationships among these individuals. As a result, we were unable to build directed networks to measure in-degree and out-degree vertices and edges. Moreover, our study required detailed and high quality de-identifiable individual-level data; it cannot be replicated with non-detailed data. Most health departments around the world cannot provide such detailed publicly available data in situations where COVID-19 incidence rates are high. For example, it is unrealistic to track and release such detailed data when thousands of individuals are infected per day. Hong Kong’s incidence rate before wave 5 was low enough to allow the health department to release detailed de-identifiable individual-level data. Secondly, the study was dependent on publicly available data, which prevented us from probing into COVID-19 cases’ contact tracing records. Thirdly, it was likely that the publicly available data may have underreported individuals’ travel records as a result of the overwhelming workload among contact tracing officers. Lastly, the lack of follow-up among some of the hard-to-track vulnerable populations was an unsolved issue in our study. As a result, our findings may underestimate the actual probability of edges formation among these vulnerable populations, especially among flight crews and sailors. Global cooperation is needed to track and follow-up these vulnerable populations to improve gaps in our understanding of the infection and access to medical care.

Future studies are encouraged to explore research questions such as whether temporal changes in health policies are associated with temporal changes in vulnerable population structures, by applying an epidemiological study design to examine potential confounders and effect modifiers. Additionally, future research could explore SARS-CoV-2 transmission dynamics among vulnerable populations and how it differs from the general population by utilizing traditional infectious disease transmission dynamics models, such as susceptible-infectious-recovered (SIR) or susceptible-exposed-infectious-recovered (SEIR) models.

## 5. Conclusions

Our findings demonstrated that vulnerable populations who need to travel due to occupation during the COVID-19 pandemic showed higher risk of exposure to SARS-CoV-2 than the general population. Specially, vulnerable populations with similar ages and occupations, especially younger domestic workers and flight crew members, were more likely to be exposed to SARS-CoV-2. Our study highlights the utility of social network analysis to provide critical information on the health risks of airborne infectious diseases to hard-to-track vulnerable populations. Awareness of the disproportional health risks borne by vulnerable populations can motivates calls for health policy makers to provide additional support and resources needed to reduce risk.

## Figures and Tables

**Figure 1 ijerph-19-07565-f001:**
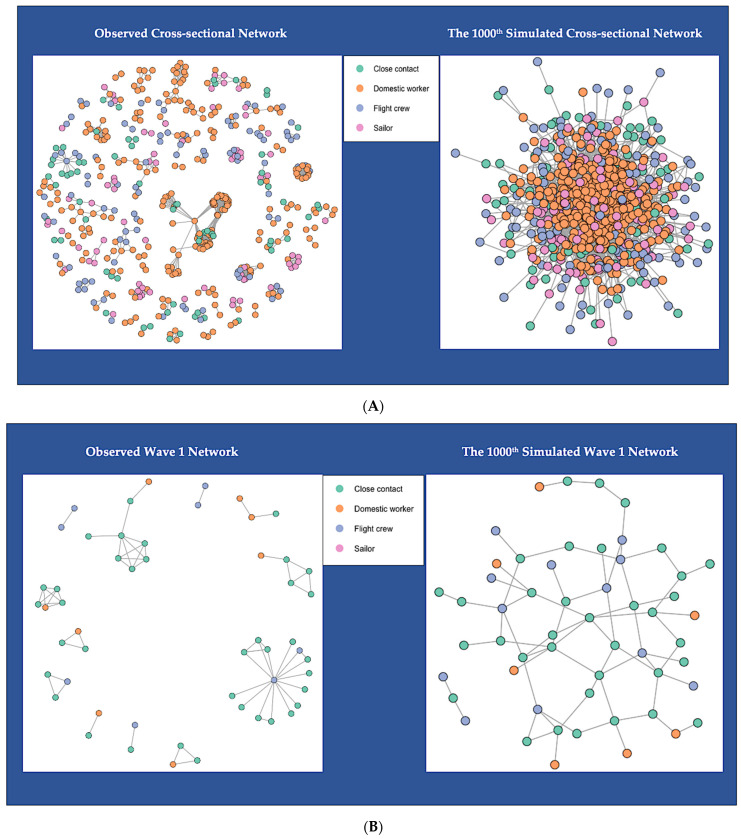
(**A**) Network Topology of Observed and Simulated Networks for Cross-sectional Cohort; (**B**) Network Topology of Observed and Simulated Networks for Wave 1 Cohort; (**C**) Network Topology of Observed and Simulated Networks for Wave 2 Cohort; (**D**) Network Topology of Observed and Simulated Networks for Wave 3 Cohort; (**E**) Network Topology of Observed and Simulated Networks for Wave 4 Cohort; (**F**) Network Topology of Observed and Simulated Networks for Wave 5 Cohort.

**Figure 2 ijerph-19-07565-f002:**
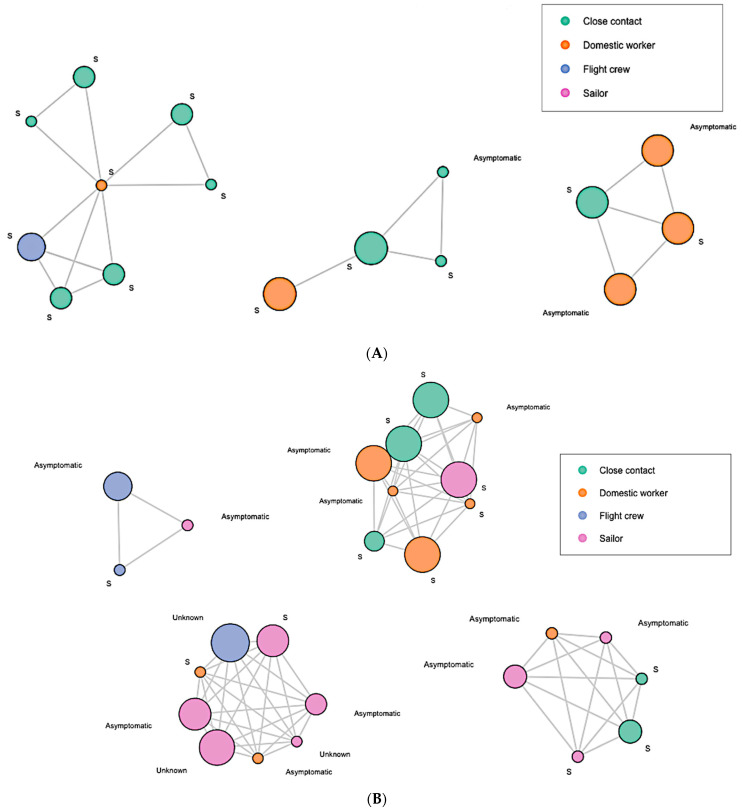
(**A**) Clusters in Wave 1 Network (Degree > 2); (**B**) Clusters in Wave 2 Network (Degree > 10); (**C**) Clusters in Wave 3 Network (Degree > 5); (**D**) Clusters in Wave 4 Network (Degree > 10); (**E**) Clusters & Large Components in Wave 5 Network.

**Figure 3 ijerph-19-07565-f003:**
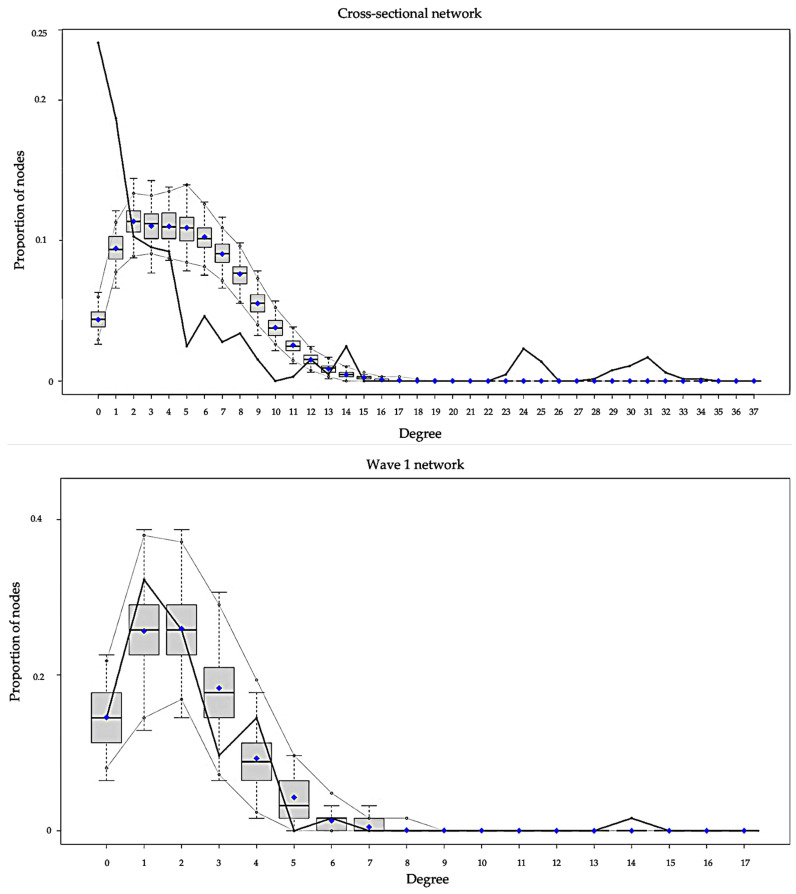
Goodness-of-fit Plots for Degree.

**Figure 4 ijerph-19-07565-f004:**
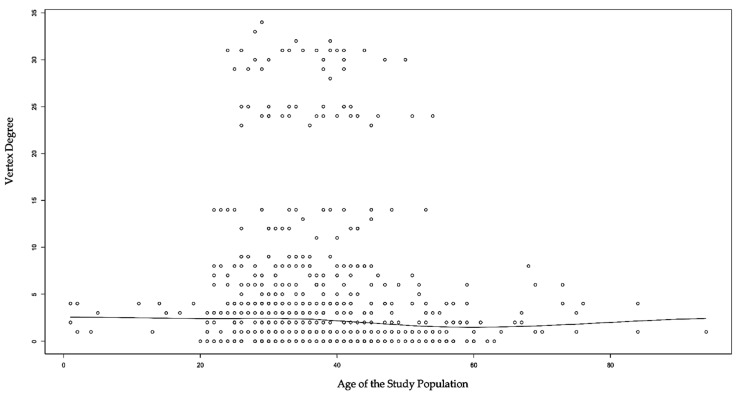
Association Between Age of the Study Population and Vertex Degree.

**Figure 5 ijerph-19-07565-f005:**
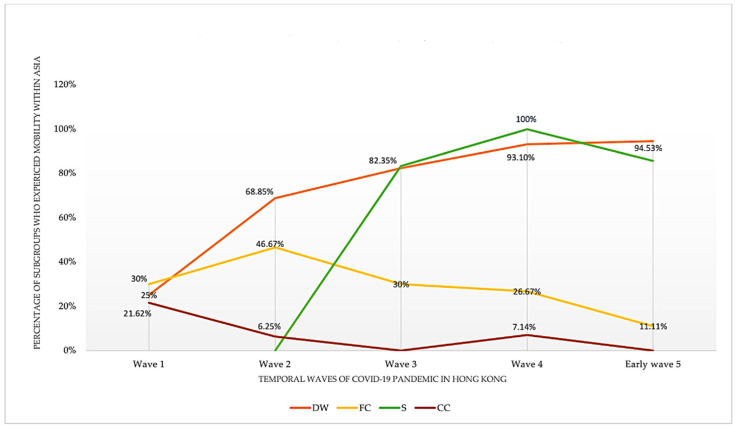
Study Population’s Temporal Mobility within Asia (non-Hong Kong Territory).

**Figure 6 ijerph-19-07565-f006:**
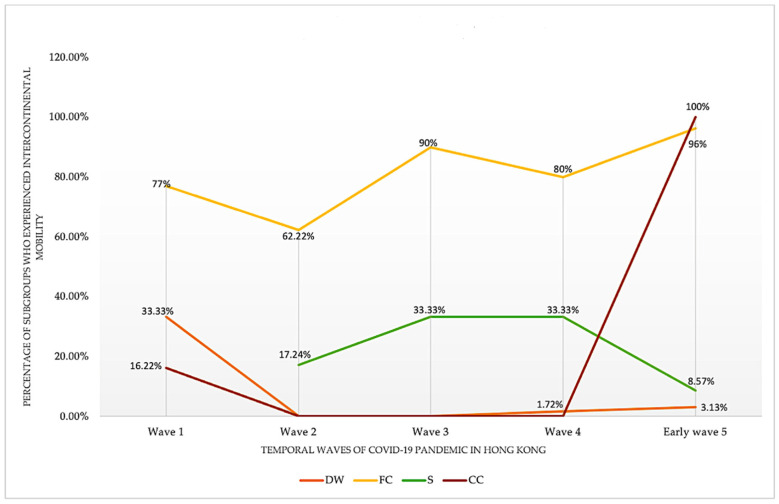
Study Population’s Temporal Intercontinental Mobility (America, Europe, Africa and Oceania).

**Table 1 ijerph-19-07565-t001:** Temporal Stratification of the Five Wave Cohorts.

Cross-Sectional Cohort (Overall)	January 2020–January 2022
Wave 1 cohort	January 2020–May 2020
Wave 2 cohort	June 2020–October 2020
Wave 3 cohort	November 2020–December 2020
Wave 4 cohort	January 2021–May 2021
Early wave 5 cohort *	June 2021–January 2022

* Wave 5 in Hong Kong started from January 2022, and not end yet at the time of writing.

**Table 2 ijerph-19-07565-t002:** Eight Scenarios for Edges Formation.

1	Cohabited in the same apartments with family members or roommates
2	Took same flights on the same dates
3	Cohabited with employer *
4	Social gathered at the same sites on the same dates
5	Worked in person with coworkers at the same physical sites on the same dates
6	Provided or received customer service in person
7	Physically presented at same departure and arrival airports within two days ** or physically presented at the same arrival airports on the same dates
8	Lived or quarantined in the same hotels within 14 days ***

* Domestic workers are required by the Hong Kong government to live-in with their employers. ** For international flights, we added one day to account for time differences. *** Hong Kong government required, 14 days quarantine for newly arrived visitors during our study period.

**Table 3 ijerph-19-07565-t003:** Descriptive Statistics of Demographic Characteristics.

	Sample Size (n=)	Mean Age (±SD) *	COVID-19 Status (% Asymptomatic)	Gender (% Female)	Study Population Category Size and Distribution (n = DW, n = FC, n = S, n = CC %DW, %FC, %S, %CC) **
Cross-sectional cohort (overall)	652	37.42 (±11.41)	65.64%	64.57%	DW = 355, FC = 120, S = 105, CC = 72 55% DW, 18% FC, 16% S, 11% CC
Wave 1 cohort	62	43.34 (±16.36)	14.52%	58.06%	DW = 12, FC = 13, CC = 37 19%DW, 21%FC, 60%CC
Wave 2 cohort	201	38.13 (±12.42)	65.17%	47.28%	DW = 82, FC = 45, S = 58, CC = 16 41% DW, 22% FC, 29% S, 8% CC
Wave 3 cohort	46	36.65 (±11.36)	67.39%	56.52%	DW = 17, FC = 20, S = 6, CC = 3 37% DW, 43% FC, 13% S, 7% CC
Wave 4 cohort	151	34.86 (±10.04)	79.47%	83.44%	DW = 116, FC = 15, S = 6, CC = 14 77% DW, 10% FC, 4% S, 9% CC
Early wave 5 cohort	192	36.94 (±8.36)	71.35%	71.88%	DW = 128, FC = 27, S = 35, CC = 2 67% DW, 14% FC, 18% S, 1% CC

* SD = standard deviation. ** DW = domestic worker, FC = flight crew, S = sailor, CC = close contact.

**Table 4 ijerph-19-07565-t004:** Descriptive Statistics of Network Characteristics.

	Network Size (Number of Vertices and Edges)	Density	Measures of Centrality			Isolates * (n=)
			Degree Mean (Min.–Max.)	Betweenness Mean (Min.–Max.)	Eigenvector Mean (Min.–Max.)	
Cross-sectional network	vertices = 652 edges =1659	0.0078	10.18 (0–68)	16.96 (0–2616.67)	0.0098 (0–0.1855)	157
Wave 1 network	vertices = 62 edges =64	0.0338	4.13 (0–28)	3.613 (0–172)	0.0537 (0–0.6103)	9
Wave 2 network	vertices = 201 edges =252	0.0125	5.02 (0–18)	3.08 (0–81.33)	0.0149 (0–0.3333)	63
Wave 3 network	vertices = 46 edges =50	0.0483	4.35 (0–12)	2.65 (0–38)	0.058 (0–0.4108)	11
Wave 4 network	vertices = 151 edges =233	0.0206	6.17 (0–26)	11.68 (0–244.67)	0.0243 (0–0.2808)	36
Early Wave 5 network	vertices = 192 edges =1060	0.0578	22.08 (0–68)	43.39 (0–2616.67)	0.0332 (0–0.1855)	38

* Isolates are isolated vertices with degree of zero that have no edges connected to other vertices.

**Table 5 ijerph-19-07565-t005:** Best-Fit-Models and Goodness-of-Fit.

	Null Models		Best-Fit-Models		
	Terms	AIC Value *	Terms	AIC Value	Goodness-of-Fit Monte Carlo Empirical *p*-Value **
Cross-sectional network	Edges	19,375	Edges + age + COVID-19 symptomatic status + vulnerable population category + gender	17,444	Edges: 0.92 Age: 0.94 Symptomatic status: 0.46–0.86 Category: 0.62–0.98 Gender: 0.86
Wave 1 network	Edges	554.5	Edges + vulnerable population category	549.8	Edges: 0.96 Category: 0.86–0.96
Wave 2 network	Edges	2701	Edges + COVID-19 symptomatic status + vulnerable population category	2451	Edges: 0.96 Symptomatic status: 0.76–0.96 Category: 0.98–1
Wave 3 network	Edges	402.6	Edges + age + vulnerable population category	384.8	Edges: 1 Age: 0.92 Category: 1
Wave 4 network	Edges	2265	Edges + COVID-19 symptomatic status + vulnerable population category + gender	2134	Edges: 1 Symptomatic status: 0.98 Category: 1 Gender: 1
Early Wave 5 network	Edges	8103	Edges + age + vulnerable population category	6967	Edges: 0.86 Age: 0.84 Category: 0.6–0.98

*** Smaller AIC value is better. ** A Monte Carlo empirical *p*-value < 0.05 indicates poor fit of the model predictor, while large Monte Carlo empirical *p*-value > 0.05 indicates good fit of the model predictor.

**Table 6 ijerph-19-07565-t006:** Study Population’s Cross-sectional Mobility.

Study Population Category (n=) *	Mobility within Asia (Non-Hong Kong Territory) (n = **, % ***)	Intercontinental Mobility (America, Europe, Africa, and Oceania) (n = **, % ***)
Domestic worker (n = 355)	300 (84.5%)	10 (2.82%)
Flight crew (n = 120)	37 (30.83%)	94 (78.33%)
Sailor (n = 105)	95 (90.48%)	17 (16.19%)
Close contact (n = 72)	10 (13.89%)	8 (11.11%)

* Sample size of subgroups. ** Number of cases experienced mobility. *** Proportion of the cases experienced mobility among the subgroups.

## Data Availability

All the data were publicly available. Daily situation reports of COVID-19 cases published by the Center for Health Protection in the Hong Kong Department of Health can be retrieved from https://www.coronavirus.gov.hk/eng/index.html (accessed on 25 January 2022). Contact tracing reports of each COVID-19 case can be retrieved from https://wars.vote4.hk/en/cases (accessed on 25 January 2022).

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
