# Peer review of "Social Network Analysis on the Mobility of Three Vulnerable Population Subgroups: Domestic Workers, Flight Crews, and Sailors during the COVID-19 Pandemic in Hong Kong"

_ijerph, 2022, doi:10.3390/ijerph19137565_

Round 1

Reviewer 1 Report

The authors hypothesized that vulnerable populations with increasing mobility will have higher risk of exposure to SARS-CoV-2. Therefore, their study aims were to explore social networks among the three vulnerable population subgroups and capture temporal changes in their probability of being exposed to SARS-CoV-2 via mobility.

To this end, the authors used the data of COVID-19 case daily reports and contact tracing records from the Center for Health Protection in Hong Kong Department of Health and employed the Exponential Random Graph Models (ERGMs) model to explore and analyze social networks.

The authors found and reported in the manuscript interesting results that deserve to be considered for publication. However, there are still some lacks and the authors should first consider the following points before consideration for publication.

General comments:

Aside from COVID-19, three key words characterize this work: vulnerable populations, social network and mobility. Therefore:

1 - The authors should consider :
    (i) - vulnerable populations: provide definition and references
    (ii) - social network: define in this context and more related references, add more references and related work using ERGMs (Refs. 28- 30 not enough)
    (iii) - mobility: missing literature

2 - The manuscrit is lacking of the considerations on transmission dynamics or epidemiological model. This should be rectified

3 - Mobility: description, characterization and analysis is missing

Specific comments

4 - Abstract: reshape, the objectives are missing

5 - Pages 119 - 122: provide definitions of Betweenness and Eigenvector centralities

6 - Pages 123 - 124: make clear whether the isolated edges (how many) are removed from the analysis

7 - Page 154: Define the Homophily

8 - Figure 1: quality is poor. Improve the visibility and lisibility

9 - Figure 9: Very poor quality. Improve the visibility and lisibility

Author Response

Thank you.

Reviewer 2 Report

Dr. Alice Yang,

Assistant Editor IJERPH

Dear Dr. Yang,

The Manuscript ID: ijerph-1728679 entitled “Social Network Analysis on Mobility of Three Vulnerable Population Subgroups: Domestic Workers, Flight Crews, and Sailors during the COVID-19 Pandemic in Hong Kong analysed data from public databases from Hong Kong in order to find the probability of exposure of tree vulnerable populations during the five different waves of the COVID-19 pandemic. The authors showed that the domestic workers were 4.56 times in wave 2 and 1.81 times in general more likely to be exposed to SARS-CoV-2 than other vulnerable populations. Also, they presented an extremely low mortality rate but high rate of asymptomatic cases among their study population. The manuscript is well written and will be of high interest to the IJERPH public and to those working with COVID-19. Find below few suggestions in order to improve the manuscript.

Specific comments:

1- Figure 1, Page 8, 9, and 10: Please, identify the figures with subheadings, such as Figure 1A, Figure 1B, Figure 1C….”. In addition, increase the font size described in the legends in the left part of the figure. The way they are presented, it is difficult to read them due to a small size.

2- Figure 3 and 4, Page 15-16: Please, the identification of the “y” axis of all the graphics of both figures are to small to read and not sharp. It would be interesting if they could be improved.

Author Response

Thank you.

Reviewer 3 Report

The authors used social network methods to analyze the mobility of three vulnerable population subgroups. It is a very interesting topic. The following are a few concerns I have for this paper:

1. The study population is very different in the 5 waves. The difference found between waves might be caused by subgroup difference. The population difference is confounded with waves. It is difficult to figure out the difference between waves is wave difference or subgroup difference. 

2. Because of the population difference of the 5 waves, it is difficult to interpret the results from cross-sectional cohort. Maybe it is better to separate the 3 subgroups in the cross-sectional cohort. Another of my concerns is why the 3 subgroups are pooled together in this study. They are not very similar. 

3. There should be much more domestic workers than flight crews in Hong Kong. The ratio in this study is 55:18 ~= 3:1. If the ratio between domestic workers and flight crews is much larger than 3:1, does that mean domestic workers are less connected. Shall we consider the population size of subgroups in the analysis? For example, there are 400,000 domestic workers in Hong Kong, only 358 are affected (0.1%), and assuming there are 2000 flights crews,  117 are affected (6%). Domestic workers are less likely to be connected. 

4. Why are close contacts included in this analysis? Especially in wave 1, there are 60% close contacts. The results are based on the jobs of close contacts instead of 3 subgroups. 

Author Response

Thank you.

Round 2

Reviewer 1 Report

The authors responded satisfactorily to all my questions and comments. Therefore, I recommend publishing this material.